# A retrospective cohort study of prescribing outcomes in outpatients treated with nirmatrelvir–Ritonavir for COVID-19 in an interdisciplinary community clinic

**Valerie Leung**[1], **Suzanne Gill**[1], **Andrea Llanes**[1], **Armughan Khawaja**[1], **Amanda Stagg**[1], **Janine McCready**[1], **Mariana Jacubovich**[1], **Grace Ho**[1], **Jeff Powis**[1], **Christopher Kandel**[1,2]*

1 Michael Garron Hospital, Toronto East Health Network, Toronto, Ontario, Canada, 2 Department of Medicine, University of Toronto, Toronto, Ontario, Canada

* Christopher.Kandel@mail.utoronto.ca

## Abstract

### Background

Large observational studies have demonstrated the real-world effectiveness of nirmatrelvir–ritonavir in preventing severe COVID-19 in higher risk individuals, but have provided limited information on other aspects of nirmatrelvir-ritonavir use. Our objective was to evaluate prescribing outcomes such as the prevalence of drug-drug interactions (DDI), adverse drug events (ADE) and treatment adherence in an outpatient community clinic setting.

### Methods

We conducted a single-centre retrospective cohort study of adult outpatients prescribed nirmatrelvir–ritonavir in our community COVID-19 assessment clinic in Toronto, Ontario between March 3 and September 20, 2022. We performed a descriptive analysis of the patient population, DDIs, DDI interventions, treatment adherence, ADEs and clinical outcomes of patients prescribed nirmatrelvir–ritonavir.

### Results

There were 637 individuals prescribed nirmatrelvir–ritonavir during the study period. The median age was 70, the median number of risk factors for severe disease were 2, 45% were immunocompromised and 82% had received 3 or more COVID-19 vaccine doses. 95% (542/572) completed the 5-day course of therapy with 68% (388/572) having complete symptom resolution by 28-day. Eleven percent (60/572) experienced recurrent symptoms following the completion of nirmatrelvir–ritonavir. Over 70% had one or more clinically significant DDIs requiring mitigation and 62% of patients experienced at least one ADE, which was most commonly dysgeusia or gastrointestinal-related. Ninety-five percent (542/572) of patients completed therapy as prescribed. Overall, hospitalization within 28 days was 3.3% with 1.2% attributed to COVID-19 and there were no deaths.

**Data Availability Statement:** Data cannot be shared publicly because of the requirements of the Michael Garron Hospital Research Ethics Board.

Data are available from the the Research Ethics Board (contact via ResearchEthicsBoard@tehn.ca) for researchers who meet the criteria for access to confidential data.

**Funding:** This project was funded by the TD Community Health Solutions Research Grant and by in-kind contributions from MGH. TD had no role in study design, data collection and analysis, decision to publish, or preparation of the manuscript.

**Competing interests:** The authors have declared that no competing interests exist.

## Interpretation

Nirmatrelvir–ritonavir was associated with a high prevalence of clinically significant DDIs, which required mitigation strategies and a high frequency of mild ADEs. Collaborative assessment to address medication alterations resulted in high treatment adherence.

## Introduction

Nirmatrelvir–ritonavir reduces the risk of hospitalization among unvaccinated individuals with COVID-19 [1]. Large observational studies using administrative databases have similarly demonstrated reductions in hospitalization with nirmatrelvir–ritonavir in vaccinated individuals, however these lack detailed data on adverse drug events (ADE) including those that do not require medical attention and potentially rare effects such as liver toxicity, medication adherence and mitigation strategies for drug-drug interactions (DDI) [2–5]. In addition, these studies are prone to misclassification when determining reason for hospitalization [6]. To complement existing studies and address the potential harms of nirmatrelvir–ritonavir detailed, individual-level clinical information is necessary.

In Ontario, Canada nirmatrelvir–ritonavir is funded for outpatients based on clinical criteria determined by the Ministry of Health, including individuals who were; immunocompromised, older than age 70, age 60 and older with fewer than 3 vaccine doses, age 18 or older with less than 3 COVID-19 vaccine doses and one or more risk conditions (obesity, diabetes, heart disease, hypertension, congestive heart failure, chronic respiratory disease including cystic fibrosis, cerebral palsy, intellectual or developmental disability, sickle cell disease, moderate or severe kidney disease, moderate or severe liver disease and pregnancy) [7]. Third doses of COVID-19 vaccines were first available in August 2021 to select vulnerable populations, expanded in September 2021 to individuals at highest risk of severe illness before being available to all adults in December 2021 [8–11]. The Ontario COVID-19 Science Advisory Table was the main source of clinical guidance on the use of COVID-19 therapeutics including dosing and other clinical aspects such as how to manage potential drug-drug interactions [12]. The primary objective of this study is to describe and quantify potential drug-drug interactions (DDI) and the associated mitigation strategies employed. The secondary objectives are to characterize treatment adherence, prevalence of adverse events and describe clinical outcomes of patients who received nirmatrelvir–ritonavir.

## Methods

### Population and setting

We included consecutive adults (age >18) who were outpatients prescribed nirmatrelvir–ritonavir between March 3 and September 20, 2022 at a community COVID-19 Clinical Assessment Centre (CAC). March 2022 was when CACs were able to provide nirmatrelvir-ritonavir and a 6-month timeframe was selected in order to capture sufficient information for meaningful analysis. The CAC clinic was operated at Michael Garron Hospital (MGH), a large community teaching hospital in Toronto, Ontario. The CAC utilized a centralized, collaborative interdisciplinary model in which all individuals presenting to the CAC, Emergency Department (ED) or a Community Outreach Centre (COC) with a positive COVID-19 molecular test were evaluated in-person or virtually by a healthcare provider.

Individuals meeting the provincial clinical criteria for treatment with nirmatrelvir-ritonavir who did not have unavoidable contraindications were evaluated by a pharmacist who performed a best-possible medication history, assessed DDIs, confirmed dosing based on renal function, counseled and dispensed the medication to the patient or arranged for home delivery. For patients without a creatinine in the previous 90 days, renal function was assessed before nirmatrelvir-ritonavir was provided. When assessing for potential DDIs with nirmatrelvir/ritonavir, pharmacists consulted a minimum of 2 standard references (i.e. Liverpool COVID-19 and one other standard reference) [13–16]. If no interaction was identified after checking with 2 references, pharmacists were encouraged to check a third reference for confirmation if there was uncertainty. Due to lack of data and as a precaution, pharmacists advised patients to hold all non-essential vitamins, minerals and natural health products while on nirmatrelvir/ritonavir. Mitigation strategies were determined in collaboration with the prescriber, communicated to the patient and documented in the electronic patient record using a standardized template. Where changes to comedications were required, patients were counselled to either hold or adjust the comedication for 7 days and then received a follow-up phone call from the pharmacist a week later reminding them to resume their usual regimen [13].

Patients were followed virtually during treatment (or until clinical resolution) by an interdisciplinary team of nurses overseen by a physician using either a remote monitoring application (Vivify Health®) and/or by telephone calls. Patients were also followed-up virtually at least 28 days following completion of therapy by a member of the study team. Residents of long-term care homes and patients that received nirmtrelvir-ritonavir while admitted as inpatients were excluded as these individuals could not be followed by the CAC. We adhered to the Strengthening the Reporting of Observational Studies in Epidemiology (STROBE) reporting guideline [17].

## Data sources & definitions

Risk factors for severe disease and the definition of immunocompromised individuals were consistent with the Ontario Science Table guidelines; immunocompromised individuals included those receiving immunosuppressive agents (e.g. for treatment of solid tumors and hematologic malignancies), those with moderate or severe primary immunodeficiency and individuals with advanced or untreated HIV infection [13]. Clinical information was abstracted from standardized templates in the electronic medical record (Cerner Powerchart). Information on adherence, ADE, and the clinical trajectory were self-reported by patients at the 28-day assessment. Patient characteristics and clinical outcomes (symptom resolution, healthcare utilization and death) were abstracted by AL and AS and prescribing outcomes (prevalence and severity of DDI, mitigation strategies, treatment adherence, ADE) were abstracted by 2 pharmacists (VL, SG) from June 6, 2022 until December 12, 2022. Data were reviewed by the study team for face validity prior to analysis. Hospitalizations and emergency department (ED) visits were reviewed by 2 physicians (JP, CK) to determine whether they were related to progression of COVID-19. When consensus could not be achieved a third physician (JM) was consulted. Authors had access to identifiable information during data collection. A DDI was defined as the concurrent use of co-medication(s) potentially resulting in decreased effectiveness and/or increased ADE from nirmatrelvir–ritonavir or the co-medication(s). Potential DDIs identified were assigned by VL and SG to a severity classification based on standard references and previous studies (i.e. level 1 –contraindicated, level 2 –clinically significant requiring mitigation, and level 3 –does not typically require mitigation) [13–16, 18, 19] (S1 Table). Since patients were advised to stop all non-essential vitamins, minerals and supplements while on treatment, these were excluded as potential DDIs.

### Statistical analysis

The population was described using proportions for categorical variables and medians with interquartile range for continuous variables. The DDIs were categorized by severity and broad medication categories. Data were analyzed using Microsoft Excel 2013 and R version 4.2.0.

### Ethics approval

This study was approved by the MGH Research Ethics Board with a waiver for informed consent as this was a retrospective cohort study.

## Results

Overall, 637 individuals were prescribed nirmatrelvir–ritonavir, with 8 patients receiving a second course for a subsequent infection. The median age was 70 (IQR 51–78) and 56% (356/637) were females. The median number of risk factors for severe disease was 2, most commonly age 70 or older (52%, 332/637) and a cardiovascular condition (50%, 316/637); 45% (285/637) were immunocompromised (Table 1). Very few patients were unvaccinated (8%, 48/637) as most had received 3 or more vaccine doses (82%, 523/637). The median time from symptom onset to treatment was 2 days (IQR 1–3) with 82% (527/645) starting treatment within 3 days. Dose reduction for renal dysfunction was required for 29% (185/645) of individuals (Table 1). Those who received a second course were younger, (median age 47) and 88% (7/8) were immunocompromised; the median number of days between courses was 110.5 (IQR: 88–128).

The median number of concurrent medications was 7 (IQR: 5–10) with 70% (450/645) of patients having at least 1 DDI, increasing to 82% (273/332) for older adults 70 years and older (Table 2). Of 840 DDIs, 95% (795/840) were categorized as level 2 clinically significant requiring intervention. The most common interacting co-medications were cardiovascular medications (55%, 460/840) with the vast majority prescribed for management of hypercholesterolemia, hypertension and benign prostatic hyperplasia (49% 411/840). The next most common categories were central nervous system (19%, 155/840) and oral antithrombotic agents including 27 direct oral anticoagulants (5%, 43/840) (S2 Table). Out of 829 mitigation strategies, the co-medication was held in 56% (467/829) with the majority of these being HMG CoA reductase inhibitors (246/467), dose adjusted in 22% (179/840) and an alternative drug started in 8% (64/840) of cases.

A virtual follow-up assessment was completed for 89% (572/645) of patients; those unable to be reached for follow-up were excluded in analysis of ADE, treatment adherence and clinical outcomes. Those lost to follow-up were younger, but otherwise similar. At follow-up, 62% (351/572) reported one or more ADE potentially related to nirmatrelvir–ritonavir, most commonly dysguesia (39%, 223/572), diarrhea (23%, 131/572) and nausea or vomiting (19%, 98/572). Despite ADEs, 95% (542/572) completed therapy as prescribed and 87% (495/572) stated that they were willing to take nirmatrelvir–ritonavir again in the future if offered (Table 2).

Most patients reported symptom resolution after treatment, however 32% (184/572) experienced lingering symptoms that was most commonly fatigue followed by cough. While 11% (60/572) experienced worsening of symptoms 7 days or less after completing treatment, most patients reported that their rebound symptoms were milder and none were severe enough to require retreatment. Within 28-days, 9% (54/572) visited the ED, 8% (47/572) visited a family physician or walk-in clinic and 3% (19/572) were hospitalized. Of the ED visits that did not require admission, 54% (19/35) were related to COVID-19 symptoms with most presenting within 7 days of starting nirmatrelvir–ritonavir. The other visits were unrelated to COVID-19 and occurred more than 10 days after staring therapy. Only 37% (7/19) of hospital admissions

**Table 1. Characteristics of study population.**

| Patient Characteristic | % (n) |
|---|---|
| | N = 637 |
| Age, years | |
| Median (IQR) | 70 (51–78) |
| < 20 y | 0.5% (3) |
| 20–39 y | 13.7% (87) |
| 40–59 y | 19.6% (125) |
| 60–69 y | 14.1% (90) |
| 70–79 y | 30.8% (196) |
| > 80 y | 21.4% (136) |
| Sex, female | 56.4% (356) |
| Number of home medications[1], Median (IQR) | 7 (5 to 10) |
| Risk Factors for Severe COVID-19[2,3] | |
| Number of risk factors, Median (IQR) | 2 (0 to 3) |
| Age > 70 | 52.1% (332) |
| Heart disease, Hypertension, Congestive Heart Failure | 49.6% (316) |
| Kidney disease (eGFR, 60 mL/min) | 27.3% (174) |
| Chronic respiratory disease | 22.8% (145) |
| Diabetes | 18.7% (119) |
| Obesity (BMI >30 kg/m2) | 13.8% (88) |
| Intellectual & developmental disabilities | 4.2% (27) |
| Liver disease (e.g., Child Pugh Class B or C cirrhosis) | 0.3% (2) |
| Cerebral palsy | 0% (0) |
| Sickle cell disease | 0% (0) |
| Immunocompromised or Immunosuppressed Individuals[4] | 44.7% (285) |
| Pregnancy | 0.002% (1) |
| Vaccine Doses | |
| 0 | 7.5% (48) |
| 1 | 1.4% (9) |
| 2 | 8.5% (54) |
| 3 | 37.0% (236) |
| 4 | 44.0% (280) |
| 5 | 1.6% (10) |
| Time from symptom onset to start of therapy[5], d | |
| Median (IQR) | 2 (1 to 3) |
| 3 days or less | 81.7% (527) |
| 4–5 days | 18.3% (118) |
| Initial Assessment[5] | |
| In-person | 77.0% (497) |
| Virtual | 23.0% (148) |
| Completed 28 day Follow-up Assessment[5] | 88.7% (572) |

[1]Home medications includes prescription medications and non-prescription medications as well over the counter products as identified by Best Possible Medication History conducted at the time of Nirmatrelvir–ritonavir prescribing

[2]as per Ontario COVID-19 Science Advisory Table criteria [10]

[3]other than being immunocompromised

[4]Based on Ontario COVID-19 Science Advisory Table definition: includes individuals receiving immunosuppressive agents, those with moderate or severe primary immunodeficiency and individuals with advanced or untreated HIV infection [10]

[5]based on N = 645 individuals as 8 received a second course for a subsequent infection.

**Table 2. Prescribing & clinical outcomes.**

| Outcome | % (n) |
|---|---|
| | N = 645 |
| Dose reduction for renal dysfunction | 28.7% (185) |
| Prevalence of DDIs (n = 840) | |
| Cardiovascular drugs | 54.8% (460) |
| Lipid modifying agents, | 29.3% (246) |
| Calcium Channel Blockers, | 13.9% (117) |
| Alpha-blockers, | 5.7% (48) |
| Phosphodiesterase Type 5 Inhibitors, | 2.0% (17) |
| Other | 3.8% (32) |
| Central Nervous System | 18.5% (155) |
| Benzodiazepine receptor agonist | 5.8% (49) |
| Antidepressants | 4.4% (37) |
| Opioids | 3.3% (28) |
| Antipsychotics | 1.7% (14) |
| Other CNS agents | 3.2% (27) |
| Oral antithrombotic agents | 5.1% (43) |
| Corticosteroid | 4.4% (37) |
| Gastrointestinal/Urinary agents | 2.9% (24) |
| Immunosuppressants | 2.9% (24) |
| Antihistamines | 2.5% (21) |
| Hormones | 2.4% (20) |
| Autonomic agents | 2.5% (21) |
| Antineoplastic agents | 2.0% (17) |
| Other | 2.1% (18) |
| DDI Severity | |
| Level 1 (Contraindicated) | 0.12% (1)[1] |
| Level 2 (Clinically Significant) | 94.6% (795) |
| Level 3 (Minor) | 5.2% (44) |
| Patients with 1 or more DDI | |
| Level 1 or 2 | 68.8% (444) |
| Any Severity | 70.1% (450) |
| Patients ≥ 70 years of age with 1 or more DDI[2] | |
| Level 1 or 2 | 81.0% (269) |
| Any Severity | 82.2% (273) |
| Mitigation Strategies (n = 829) | |
| Co-medication held | 56.3% (467) |
| Dose of co-medication adjusted | 21.6% (179) |
| Alternative drug initiated | 7.7% (64) |
| Co-medication continued with additional monitoring | 12.4% (103) |
| Other | 1.9% (16) |
| Treatment adherence to 5 day course of therapy[3] | 94.8% (542) |
| Adverse Events (ADE)[3] | |
| Patients reporting ≥1 ADE | 61.4% (351) |
| Dysgeusia | 39.0% (223) |
| Diarrhea | 22.9% (131) |
| Nausea | 12.2% (70) |
| Vomiting | 4.9% (28) |

(*Continued*)

**Table 2.** (Continued)

| Outcome | % (n) |
|---|---|
| | N = 645 |
| Acid Reflux | 1.2% (7) |
| Dyspepsia | 1.0% (6) |
| Other | 6.6% (38) |
| ADE resulting in discontinuation of therapy | 4.2% (24) |
| Complete symptom resolution after completion of treatment[3] | 67.8% (388) |
| Rebound symptoms[4] | 10.5% (60) |
| Healthcare utilization[3] | |
| ED visit | 9.4% (54) |
| Family MD or walk-in clinic visit | 8.2% (47) |
| Hospital admission | 3.3% (19) |
| Progression of symptoms | 0.9% (5) |
| Adverse event related to therapy | 0.3% (2) |
| Unrelated to COVID-19/therapy | 2.1% (12) |
| Death | 0% (0) |

[1] one level 1 DDI was identified however the co-medication (lorlatinib) is only contraindicated when taken in the last 14 days(10)

[2] based on N = 332 individuals 70 or older

[3] based N = 572 individuals that could be reached for 28 day follow-up assessment

[4] defined as worsening of symptoms ≤7 days after completing treatment

were related to COVID-19; of these, 5 were due to progression of COVID-19 symptoms (including 1 patient with rebound). Two individuals were admitted to hospital after experiencing severe gastrointestinal symptoms leading to impaired oral intake (S3 Table). No deaths occurred.

## Discussion

A retrospective cohort study of 637 outpatients prescribed nirmatrelvir–ritonavir when BA.1/2 and BA.4/5 were the prevailing variants found a high prevalence of DDIs that required intervention. Nearly 70% of patients had more than one clinically significant DDI. This increased to 82% in older adults, which is similar to studies using administrative databases which found that 2/3 of older adults had ≥ 1 potential DDI [5, 20]. Almost all DDIs (98.6%) required pharmacist intervention, which likely contributed to the low rate of severe adverse events in our cohort. While a formal time-motion study was not conducted as part of this study, pharmacists typically spent 30–60 minutes per patient performing assessment, counselling and documentation activities. However, patients requiring comedication dosage changes and/or initiation of new medications while on therapy often required more attention especially in instances where changes to compliance packaging were required (i.e. blister packs). Additionally, individuals on immunosuppressive drugs (e.g. sirolimus, tacrolimus) may require therapeutic drug monitoring to guide dose adjustment following nirmatrelvir-ritonavir therapy which requires additional coordination [13].

The high frequency of ADEs (62%) did not translate into discontinuation of therapy as 95% of patients were adherent to therapy. Dysgeusia and diarrhea were most frequent at 39% and 23% respectively, which was much higher than the 6% for dysgeusia, 3% for diarrhea, and 1% for nausea observed in EPIC-HR. The higher rates in this cohort may be due to a combination

of our cohort being older, more comorbid and biases related to retrospective data collection when collecting specific ADEs [1, 21]. Nevertheless, while the real-world gastrointestinal ADEs are more prominent than originally reported, they rarely led to severe ADE and hospitalization.

By 28 days following treatment, 32% (184/572) reported persistent COVID-19 symptoms. Studies conducted before the Omicron variant appeared estimated that 10–35% of individuals have symptoms that persist for at least 28 days post COVID-19 infection [22, 23]. While we were unable to perform longer term assessments, the prevalence of lingering symptoms at 28 days appears similar to those seen with long COVID. However, without a comparison group it is not possible to determine whether nirmatrelvir–ritonavir led to a faster resolution of symptoms or a reduction in long COVID. Consistent with other studies, 11% experienced rebound symptoms [24–26]. In our study, no patients with rebound were severe enough to require retreatment however, one individual required hospitalization for progression of COVID-19 symptoms.

Overall, 3% of the cohort were hospitalized within 28 days with 1% deemed related to COVID-19 progression and there were no known deaths. A recent population-based cohort study in Ontario found that individuals dispensed nirmatrelvir–ritonavir from a community pharmacy had a rate of hospitalization from COVID-19 or all cause death rate within 30 days of 2.1% as compared to 3.7% for those that did not receive nirmatrelvir–ritonavir, which is higher than that observed in our cohort. It is important to note, however, that >30% of patients in the provincial cohort were long-term care residents, which are at higher risk of COVID-19 progression and were not included in this cohort [5]. Our rate of COVID-19-related hospitalization, however, was higher than that seen in EPIC-HR (1.2% vs 0.8%) likely related to differences in patient populations.

## Limitations

A limitation of our study is the lack of a comparison group that would permit a comparison of our collaborative model to alternative models of nirmatrelvir–ritonavir provision. Nevertheless, we believe that the high prevalence of clinically significant DDIs and the required mitigation strategies are generalizable beyond our single centre given the numerous known DDIs associated with nirmatrelvir–ritonavir [20]. We also did not have complete follow-up resulting in potential survivor bias if those experiencing negative outcomes were less able to participate in re-assessment; however, the 11% lost to follow-up were younger and otherwise similar with respect to vaccination status and immunosuppression status. Data on outcomes were predominately determined at 28-day assessment and were self-reported potentially leading to recall bias. A major strength is our collection of detailed clinical information to assess real-world prevalence, severity and mitigation of DDIs, adherence and ADEs not requiring healthcare visits, which reinforces the need for health care providers to perform detailed assessment and management of DDIs for most patients prescribed nirmatrelvir-ritonavir.

## Interpretation

A retrospective cohort of 637 patients found that the use of nirmatrelvir–ritonavir is frequently accompanied by clinically significant DDIs that require mitigation. With a multidisciplinary care model, there was excellent adherence and no severe outcomes related to DDIs. Individuals in our study also experienced high rates of ADE, mostly dysgeusia and gastrointestinal, but these were predominantly minor and only 2 individuals were hospitalized. In general, we found that nirmatrelvir-ritonavir is well tolerated; almost all patients completed therapy and most were willing to take the medication again if offered.

As the pandemic continues to evolve, the use of nirmatrelvir-ritonavir has become a mainstay of therapy for outpatient treatment of mild COVID-19 in individuals at high risk of disease progression. The significance of our findings reinforces the importance of managing DDIs for those prescribed nirmatrelvir-ritonavir. Prescribers should be aware of the high real-world prevalence of DDIs and ADEs associated with this therapy and be familiar with mitigation strategies to prevent harm related to DDIs and help support adherence to therapy.

## Supporting information

**S1 Table. Potential drug-drug interactions (DDI) with nirmatrelvir/ritonavir.**
(DOCX)

**S2 Table. Prevalence of DDIs.**
(DOCX)

**S3 Table. Characteristics of patients hospitalized for COVID-19.**
(DOCX)

## Author Contributions

**Conceptualization:** Valerie Leung, Suzanne Gill, Andrea Llanes, Armughan Khawaja, Amanda Stagg, Janine McCready, Mariana Jacubovich, Grace Ho, Jeff Powis, Christopher Kandel.

**Data curation:** Valerie Leung, Suzanne Gill, Andrea Llanes, Amanda Stagg.

**Formal analysis:** Valerie Leung, Suzanne Gill, Armughan Khawaja, Amanda Stagg, Jeff Powis, Christopher Kandel.

**Funding acquisition:** Suzanne Gill.

**Methodology:** Valerie Leung.

**Validation:** Valerie Leung, Suzanne Gill, Jeff Powis, Christopher Kandel.

**Writing – original draft:** Valerie Leung, Suzanne Gill, Amanda Stagg, Jeff Powis, Christopher Kandel.

**Writing – review & editing:** Valerie Leung, Suzanne Gill, Andrea Llanes, Armughan Khawaja, Amanda Stagg, Janine McCready, Mariana Jacubovich, Grace Ho, Jeff Powis, Christopher Kandel.

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
