## [Decision Letter · Decision Letter 0]

24 Aug 2023

PONE-D-23-18794A Retrospective Cohort Study of Prescribing Outcomes in outpatients treated with Nirmatrelvir–ritonavir for COVID-19 in an Interdisciplinary Community ClinicPLOS ONE

Dear Dr. Kandel,

Thank you for submitting your manuscript to PLOS ONE. After careful consideration, we feel that it has merit but does not fully meet PLOS ONE’s publication criteria as it currently stands. Therefore, we invite you to submit a revised version of the manuscript that addresses the points raised during the review process.

We look forward to receiving your revised manuscript.

Kind regards,

Victor Daniel Miron

Academic Editor

PLOS ONE

Journal Requirements:

2. Please note that in order to use the direct billing option the corresponding author must be affiliated with the chosen institute. Please either amend your manuscript to change the affiliation or corresponding author, or email us at plosone@plos.org with a request to remove this option.

Reviewers' comments:

Reviewer's Responses to Questions

**Comments to the Author**

1. Is the manuscript technically sound, and do the data support the conclusions?

Reviewer #1: Yes

Reviewer #2: Yes

Reviewer #3: Yes

2. Has the statistical analysis been performed appropriately and rigorously? 

Reviewer #1: N/A

Reviewer #2: Yes

Reviewer #3: Yes

3. Have the authors made all data underlying the findings in their manuscript fully available?

Reviewer #1: Yes

Reviewer #2: Yes

Reviewer #3: Yes

4. Is the manuscript presented in an intelligible fashion and written in standard English?

Reviewer #1: Yes

Reviewer #2: Yes

Reviewer #3: Yes

5. Review Comments to the Author

Reviewer #1: I thank the authors for sharing their experience using nirmatrelvir/ritonavir in a large cohort of patients. The finding that 70% of patients would experience a drug-drug interaction is alarming. I have a few comments to strengthen the manuscript.

1. I advise the authors to expound further on "how" the management actually took place. Once a DDI is identified and a medication adjusted there needs to be some additional follow up either internally or externally to ensure that the former dose/schedule is re-instated. Who on your team was following through?

2. Were these all consecutive patients meeting criteria? In other words, there were no other patients that were immediately not considered a candidate for NR that were excluded? When dealing with complex drug interactions that require more frequent monitoring there is always a risk that patients may not follow your instructions explicitly. Please expound on patient selection.

3. How long were medications "held" or dose-adjusted? The half-life of ritonavir is short, but the irreversible inhibition can mean that inhibition of CYP 3A4 or other may be for as long as 7-14 days. Were medications held for the duration of NR or for longer?

Reviewer #2: The paper is well written and provides information on drug-drug interactions to nirmatrelvir/ritonavir treatment on a fair number of patients. There are quite a number of DDIs, as has been reported already, most of them clinically significant. There were changes in co-medication, allowing patients to continue treatment, therefore high adherence to treatment was noted in this sample of patients. Moreover, there were no deaths.DDIs with statins were already reported in early 2022 and some protocols have requested discontinuation of statins before the initiation of treatment with nirmatrelvir/ritonavir. Is this something that had been considered?

The tables are quite informative. On table S3, in the supplement, it would it be interesting to provide info on the day treatment was initiated. Had these patients began treatment early or late post symptom initiation and diagnosis? Of course, this is just a few patients so no safe conclusions can be drawn. The study is retrospective, however, the conclusions are relevant and the strengths and weaknesses of the study are well described and acknowledged.

Reviewer #3: I want to thank the authors for this very interesting and well-wirtten paper, which analyze the clinical use of Nirmatrelvir/ritonavir by focusing on identifying and quantifying the DDI and ADE when prescribed as early treatment for COVID-19. The potential burden related to adverse events is well known in the clinical practice; however, very little literature is available on this side. Moreover, the methodology is very straightforward and accurate and limitations correctly outlined.

Some minor comments are needed in my opinion to further improve the quality of the paper.

1. I would specify whether a written informed consent was needed to be signed by participant in order to use data

2. In case of need for dose adjustment according to renal function, was the decision based on patient history (i.e known history of renal failure) or was creatinin assessed ad hoc?

3. A proper assessment of potential DDI is of outmost importance to increase adherence and therefore ensure optimal efficacy. I would consider to add a couple of sentences in the discussion on the added value of TDM for immunuppressive drugs.

4. Little literature describes also, even if rare, liver toxicity after the use of nirmatrelvir/ritonavir. Probably this aspect should also be mentioned as potential noticeable ADE that is independent from self-reporting.

6. PLOS authors have the option to publish the peer review history of their article (what does this mean?). If published, this will include your full peer review and any attached files.

Reviewer #1: No

Reviewer #2: No

Reviewer #3: No

---

## [Author Response · Author response to Decision Letter 0]

13 Sep 2023

PONE-D-23-18794

Re: A Retrospective Cohort Study of Prescribing Outcomes in outpatients treated with Nirmatrelvir–ritonavir for COVID-19 in an Interdisciplinary Community Clinic

September 12, 2023

Dear Editorial Staff and Reviewers,

Thank you for your thorough review and constructive comments. Our team has updated the manuscript based on suggestions from the reviewers and believe that the manuscript has been strengthened. 

All authors have agreed to these manuscript revisions. 

Below is the point-by-point response to each comment in bold.

Should you have any questions or additional comments, please let me know.

Sincerely,

Chris Kandel

On behalf of the study authors

Reviewer's Responses to Questions

Reviewer #1: I thank the authors for sharing their experience using nirmatrelvir/ritonavir in a large cohort of patients. The finding that 70% of patients would experience a drug-drug interaction is alarming. I have a few comments to strengthen the manuscript.

1. I advise the authors to expound further on "how" the management actually took place. Once a DDI is identified and a medication adjusted there needs to be some additional follow up either internally or externally to ensure that the former dose/schedule is re-instated. Who on your team was following through?

RESPONSE: A follow up phone call from a pharmacist affiliated with the hospital occurred on day 7. The following details have been added to the methods: Where changes to comedications were required, patients were counselled to either hold or adjust the comedication for 7 days and then received a follow-up phone call a week later from the pharmacist reminding them to resume their usual regimen. 

2. Were these all consecutive patients meeting criteria? In other words, there were no other patients that were immediately not considered a candidate for NR that were excluded? When dealing with complex drug interactions that require more frequent monitoring there is always a risk that patients may not follow your instructions explicitly. Please expound on patient selection.

RESPONSE: We included all consecutive patients who were prescribed nirmatrelvir-ritonavir by a health care provider. This strategy excluded individuals who were eligible nirmatrelvir-ritonavir, but declined the medication or were not able to safely take it on account of unavoidable drug interactions. This has now been added to the Methods section. 

3. How long were medications "held" or dose-adjusted? The half-life of ritonavir is short, but the irreversible inhibition can mean that inhibition of CYP 3A4 or other may be for as long as 7-14 days. Were medications held for the duration of NR or for longer?

RESPONSE: In general, comedications were held or dose-adjusted for the duration therapy plus an additional 2 days as per the Ontario Science Table treatment guidelines. The following details have been added to the methods: Where changes to comedications were required, patients were counselled to either hold or adjust the comedication for 7 days and then received a follow-up phone call a week later from the pharmacist reminding them to resume their usual regimen.

Reviewer #2: The paper is well written and provides information on drug-drug interactions to nirmatrelvir/ritonavir treatment on a fair number of patients. There are quite a number of DDIs, as has been reported already, most of them clinically significant. There were changes in co-medication, allowing patients to continue treatment, therefore high adherence to treatment was noted in this sample of patients. Moreover, there were no deaths. DDIs with statins were already reported in early 2022 and some protocols have requested discontinuation of statins before the initiation of treatment with nirmatrelvir/ritonavir. Is this something that had been considered? 

RESPONSE: We explicitly asked each individual prescribed nirmatrelvir-ritonavir about the use of a statin due to the concern of a DDI. Although some guidelines suggest reduction of statin dose during therapy and for 2 days after as an option, all patients on statins in this cohort study were counselled to hold their statin for the duration of treatment and an additional 2 days given the low risk of harm with temporary statin cessation. The following detail has been added to the results: Out of 829 mitigation strategies, the co-medication was held in 56% (467/829) with the majority of these being HMG CoA reductase inhibitors (246/467), dose adjusted in 22% (179/840) and an alternative drug started in 8% (64/840) of cases.

The tables are quite informative. On table S3, in the supplement, it would it be interesting to provide info on the day treatment was initiated. Had these patients began treatment early or late post symptom initiation and diagnosis? Of course, this is just a few patients so no safe conclusions can be drawn. The study is retrospective, however, the conclusions are relevant and the strengths and weaknesses of the study are well described and acknowledged.

RESPONSE: Time from symptom onset to start of therapy has been added to table S3.

Reviewer #3: I want to thank the authors for this very interesting and well-wirtten paper, which analyze the clinical use of Nirmatrelvir/ritonavir by focusing on identifying and quantifying the DDI and ADE when prescribed as early treatment for COVID-19. The potential burden related to adverse events is well known in the clinical practice; however, very little literature is available on this side. Moreover, the methodology is very straightforward and accurate and limitations correctly outlined.

Some minor comments are needed in my opinion to further improve the quality of the paper.

1. I would specify whether a written informed consent was needed to be signed by participant in order to use data

RESPONSE: As this study was a retrospective cohort a waiver of informed consent was granted by the Research Ethics Board. This has been added to the Methods section. 

2. In case of need for dose adjustment according to renal function, was the decision based on patient history (i.e known history of renal failure) or was creatinine assessed ad hoc?

RESPONSE: If a recent creatinine (<90 days) was unavailable, a creatinine was obtained in the emergency department or clinical assessment center. The following has been added under the Methods section to address this; For patients without a recent result, a creatinine level was ordered to assess renal function. 

3. A proper assessment of potential DDI is of outmost importance to increase adherence and therefore ensure optimal efficacy. I would consider to add a couple of sentences in the discussion on the added value of TDM for immunuppressive drugs.

RESPONSE: The following has been added to the discussion section: Additionally, individuals on immunosuppressive drugs (e.g. sirolimus, tacrolimus) may require therapeutic drug monitoring to guide dose adjustment following nirmatrelvir-ritonavir therapy which requires additional coordination.

4. Little literature describes also, even if rare, liver toxicity after the use of nirmatrelvir/ritonavir. Probably this aspect should also be mentioned as potential noticeable ADE that is independent from self-reporting.

RESPONSE: We agree and this was the motivation for a conducting a detailed assessment of the real-world use of nirmatrelvir–ritonavir. The following has been added to the introduction section: Large observational studies using administrative databases have similarly demonstrated reductions in hospitalization with nirmatrelvir–ritonavir in vaccinated individuals, however these lack detailed data on adverse drug events (ADE) including those that do not require medical attention and potentially rare events such as liver toxicity, medication adherence and mitigation strategies for drug-drug interactions (DDI)

---

## [Decision Letter · Decision Letter 1]

10 Oct 2023

A Retrospective Cohort Study of Prescribing Outcomes in outpatients treated with Nirmatrelvir–ritonavir for COVID-19 in an Interdisciplinary Community Clinic

PONE-D-23-18794R1

Dear Dr. Kandel,

We’re pleased to inform you that your manuscript has been judged scientifically suitable for publication and will be formally accepted for publication once it meets all outstanding technical requirements.

Kind regards,

Victor Daniel Miron

Academic Editor

PLOS ONE

Reviewers' comments:

Reviewer's Responses to Questions

**Comments to the Author**

1. If the authors have adequately addressed your comments raised in a previous round of review and you feel that this manuscript is now acceptable for publication, you may indicate that here to bypass the “Comments to the Author” section, enter your conflict of interest statement in the “Confidential to Editor” section, and submit your "Accept" recommendation.

Reviewer #1: All comments have been addressed

Reviewer #2: (No Response)

2. Is the manuscript technically sound, and do the data support the conclusions?

Reviewer #1: Yes

Reviewer #2: Yes

3. Has the statistical analysis been performed appropriately and rigorously? 

Reviewer #1: Yes

Reviewer #2: Yes

4. Have the authors made all data underlying the findings in their manuscript fully available?

Reviewer #1: Yes

Reviewer #2: Yes

5. Is the manuscript presented in an intelligible fashion and written in standard English?

Reviewer #1: Yes

Reviewer #2: Yes

6. Review Comments to the Author

Reviewer #1: Thank you for your revisions. The author group has satisfactory addressed my comments. No further revisions requested at this time.

Reviewer #2: This is a review of thr revised manusctipt. As I initially replied, the paper is well written and provides information on drug-drug interactions to nirmatrelvir/ritonavir treatment on a reasonable number of patients. There are quite a number of DDIs, as has been reported already, most of them clinically significant though. However, there have been no deaths. There were changes in co-medication, allowing patients to continue treatment, therefore high adherence to treatment was noted in this sample of patients. Other reviewers have requested further information on these changes and it is now provided. My question on whether discontinuation of statins before the initiation of treatment with nirmatrelvir/ritonavir had been considered has now been answered as has my request for info on whether these patients had began treatment early or late post symptom initiation and diagnosis. The study is retrospective, however, the conclusions are relevant and the strengths and weaknesses of the study are well described and acknowledged.

7. PLOS authors have the option to publish the peer review history of their article (what does this mean?). If published, this will include your full peer review and any attached files.

Reviewer #1: No

Reviewer #2: No

---

## [Editor Report · Acceptance letter]

12 Oct 2023

PONE-D-23-18794R1 

A Retrospective Cohort Study of Prescribing Outcomes in outpatients treated with Nirmatrelvir–ritonavir for COVID-19 in an Interdisciplinary Community Clinic 

Dear Dr. Kandel:

I'm pleased to inform you that your manuscript has been deemed suitable for publication in PLOS ONE. Congratulations! Your manuscript is now with our production department. 

Kind regards, 

on behalf of

Dr. Victor Daniel Miron 

Academic Editor

PLOS ONE